# RASCL: Rapid Assessment of Selection in CLades through molecular sequence analysis

Alexander G. Lucaci[1]*, Jordan D. Zehr[1], Stephen D. Shank[1], Dave Bouvier[2], Alexander Ostrovsky[3], Han Mei[2], Anton Nekrutenko[2], Darren P. Martin[4], Sergei L. Kosakovsky Pond[1]*

1 Institute for Genomics and Evolutionary Medicine, Temple University, Philadelphia, Pennsylvania, United States of America, 2 Department of Biochemistry and Molecular Biology, The Pennsylvania State University, University Park, PA, United States of America, 3 Krieger School of Arts and Sciences, Johns Hopkins University, Baltimore, MD, United States of America, 4 Division of Computational Biology, Department of Integrative Biomedical Sciences, Institute of Infectious Diseases and Molecular Medicine, University of Cape Town, Cape Town, South Africa

* alexander.lucaci@temple.edu (AGL); spond@temple.edu (SLKP)

## Abstract

An important unmet need revealed by the COVID-19 pandemic is the near-real-time identification of potentially fitness-altering mutations within rapidly growing SARS-CoV-2 lineages. Although powerful molecular sequence analysis methods are available to detect and characterize patterns of natural selection within modestly sized gene-sequence datasets, the computational complexity of these methods and their sensitivity to sequencing errors render them effectively inapplicable in large-scale genomic surveillance contexts. Motivated by the need to analyze new lineage evolution in near-real time using large numbers of genomes, we developed the Rapid Assessment of Selection within CLades (RASCL) pipeline. RASCL applies state of the art phylogenetic comparative methods to evaluate selective processes acting at individual codon sites and across whole genes. RASCL is scalable and produces automatically updated regular lineage-specific selection analysis reports: even for lineages that include tens or hundreds of thousands of sampled genome sequences. Key to this performance is (i) generation of automatically subsampled high quality datasets of gene/ORF sequences drawn from a selected "query" viral lineage; (ii) contextualization of these query sequences in codon alignments that include high-quality "background" sequences representative of global SARS-CoV-2 diversity; and (iii) the extensive parallelization of a suite of computationally intensive selection analysis tests. Within hours of being deployed to analyze a novel rapidly growing lineage of interest, RASCL will begin yielding JavaScript Object Notation (JSON)-formatted reports that can be either imported into third-party analysis software or explored in standard web-browsers using the premade RASCL interactive data visualization dashboard. By enabling the rapid detection of genome sites evolving under different selective regimes, RASCL is well-suited for near-real-time monitoring of the population-level selective processes that will likely underlie the emergence of future variants of concern in measurably evolving pathogens with extensive genomic surveillance.

**Data Availability Statement:** The RASCL application, depicted at high level in Fig 1, is implemented: • As a standalone pipeline (https://github.com/veg/RASCL) in Snakemake [37]. • As a

web application (https://galaxy.hyphy.org/u/hyphy/w/rapid-assessment-of-selection-on-clades-and-lineages), integrated as a workflow in the Galaxy framework, that is freely available for use on powerful public computing infrastructure (https://usegalaxy.org).

**Funding:** DPM is funded by the Wellcome Trust (222574/Z/21/Z). This research was supported in part by grants R01 AI134384 (NIH/NIAID) and grant 2027196 (NSF/DBI,BIO) to AN and SLKP. The funding bodies played no role in the design of the study, the collection, analysis, and interpretation of data, nor in writing the manuscript.

**Competing interests:** The authors have declared that no competing interests exist

**Abbreviations:** BUSTED[S], Branch-site Unrestricted Statistical Test for Episodic Diversification with synonymous rate variation; BGM, Bayesian Graphical Models; CFEL, Contrast-FEL; FADE, A FUBAR* Approach to Directional Selection (A *Fast, Unconstrained Bayesian AppRoximation for Inferring Selection); FEL, Fixed Effects Likelihood; HyPhy, Hypothesis Testing using Phylogenies; RASCL, Rapid Assessment of Selection within CLades; NCBI, National Center for Biotechnology Information; MEME, Mixed Effects Model of Evolution; TN93, Tamura-Nei, 1993; RELAX, Relaxation of selective strength; SLAC, Single-Likelihood Ancestor Counting; ViPR, Virus Pathogen Database and Analysis Resource; VOC, Variants of concern; VOI, Variants of interest.

# Introduction

Rapid characterization and assessment of clade-specific mutations that are found in persistent or rapidly expanding SARS-CoV-2 lineages have become an important component of efforts to monitor and manage the COVID19 pandemic. Identifying the most relevant mutations, e.g., those likely to impact transmission or immune escape, is a priority when new lineages are discovered, as these are key to assessing a lineage's potential threat-level. If observed mutations have not been previously characterized, computational and laboratory-based analytical approaches to inferring whether the mutations provide transmission or immune escape advantages are generally too slow to inform early public health responses.

Epidemiologically relevant mutations are likely subject to natural selection because they provide a fitness advantage [1]. Such mutations can be identified by detecting the subtle patterns of nucleotide variation within gene-sequence datasets that are indicative of selective processes. There are a multitude of powerful computational techniques that, given sufficiently informative sequence data, can identify individual codons within genes that are evolving under a range of different selective regimes [2]. Hundreds of papers and preprints have used some of the methods implemented in HyPhy [3] and Datamonkey [4] for SARS-CoV-2 selection analyses e.g., [5–8], but on small, commonly hand-curated, datasets. This is because codon-based selection analyses do not scale well to more than a few hundred sequences when using the generic out-of-the-box versions of these analyses, and because noisy sequencing data (errors in assembled consensus genomes) can drive false positives.

We developed RASCL to standardize and accelerate comparative selection detection analyses of SARS-CoV-2 variants of interest (VOI) or variants of concern (VOC). The tool has been used to study selective forces which, at least in-part, drove the emergence of the Alpha, Beta, Gamma, and Omicron VOCs [9–12]. More broadly, a tool like RASCL enables near-real-time monitoring of emergent lineages, which in turn can be used both to detect potentially adaptive mutations before they rise to high frequencies, and to help establish relationships between individual mutations and key viral characteristics including pathogenicity, transmissibility, immune evasiveness and drug resistance [13–17]. Through routine analysis of patterns of ongoing selection within individual major lineages, we can reveal the variants or circulating sub-lineages that carry potentially concerning fitness-enhancing mutations, and which would therefore most likely drive future viral transmission [18].

# Materials and methods

## RASCL application overview

The "query" set of whole genome sequences is compared against a diverse set of "background" sequences, chosen to represent globally circulating SARS-CoV-2 sequences (throughout the pandemic), and the query data set is the set of sequences which are the target of selection analyses (e.g., BA.5 clade sequences). Our background dataset is available at https://github.com/veg/RASCL/tree/main/data/ReferenceSetViPR and was assembled from Virus Pathogen Database and Analysis Resource (ViPR, viprbrc.org) [19], a curated database of publicly available viral pathogen sequences, assemblies, and genome annotations (S1 File). The inclusion of the background dataset also provides an "outgroup" clade, enabling the study of selection on branches basal to the clade of interest (COI), and to discover regions under selection pressure which are unique to the COI. The application uses several open-source tools, as well as selection analysis modules from the HyPhy software package and assembles the results from the analysis into JSON files, which can then be visualized with our full-featured ObservableHQ [20] notebook.

## Map and compress

Specifically, RASCL takes as input (i) a "query" dataset comprising a single FASTA file containing unaligned SARS-CoV-2 full or partial genomes belonging to a clade of interest (e.g. all sequences from the PANGO [17, 18] lineage B.1.617.2) and (ii) a generic "background" dataset that might comprise, for example, a set of sequences that are representative of global SARS-CoV-2 genomic diversity, e.g. those assembled from ViPR. It is not necessary to remove sequences in the query dataset that are duplicated in the background dataset—the pipeline will do this automatically.

The choice of query and background datasets is analysis-specific. For example, if another clade of interest is provided as a background, it is possible to directly identify the sites that are evolving differentially between the two clades. Other sensible choices of query sequences might be sequences from a specific country/region, or sequences sampled during a particular time-period. Note that the analysis does not require the two sets to be reciprocally monophyletic, but in many applications, this will be the case. Following the automated mapping of whole genome datasets into individual coding sequences (based on the NCBI reference annotation), the gene datasets (each containing a set of query and background sequences) are processed in parallel.

## Prepare for selection analysis

Using complete linkage distance clustering implemented in the TN93 package (tn93-cluster tool, https://github.com/veg/tn93), RASCL subsamples from available sequences while maintaining overall genomic diversity; the clustering threshold distance is chosen automatically to include no more than a user-specified number of genomes "D" (e.g., 300). In the Results section below we discuss our recent analyses of several SARS-CoV-2 clades and while other available subsampling methods for SARS-CoV-2 genomes exist which rely on spatiotemporal distributions [21], our method relies on increasing sequence diversity to enhance evolutionary signal for downstream selection analyses, while reducing computational complexity. Core method implementations in HyPhy can handle up to 25,000 subsampled sequences in a reasonable time, but the computational cost increases rapidly, and for faster turnaround 1000 sequences are the recommended setting. Following dataset compression, RASCL creates a combined (query and background) alignment with only the sequences that are divergent enough to be useful for subsequent selection analyses. Inference of a maximum likelihood phylogenetic tree with RAxML-NG, [22], or IQ-TREE, [23] is performed on the merged dataset and the query and background branches of this tree are labeled as Query or Background; internal branches of the tree are labeled using maximum parsimony.

## Selection analyses

Selection analyses are performed with state-of-the-art molecular evolution [24] models implemented in HyPhy. To partially mitigate the potentially confounding influences of within-host evolution [25, 26], where mutations occurring within an individual have not been filtered by selection at the broader population-level, and sequencing errors, selection analyses are performed only on the internal branches of phylogenetic trees, where at least one or more rounds of virus transmission are captured [27]. The following individual selection tests are applied to each gene-level alignment of merged query and background sequences.

- Branch-site Unrestricted Statistical Test for Episodic Diversification with Synonymous rate variation (BUSTED[S]): this method tests for gene-wide selection which is either pervasive

(occurring throughout the evolutionary tree) or episodic (occurring only on some lineages). [28].

- Single-Likelihood Ancestor Counting (SLAC), uses a combination of maximum-likelihood and counting approaches to infer pervasive selection through nonsynonymous (dN) and synonymous (dS) substitution rates on a per-site basis for a given coding alignment and corresponding phylogeny. We use the results from SLAC to create substitution mapping of genomic sites and selection analysis results across methods [29].

- Coevolution detection using Bayesian Graphical Models (BGM): this method identifies groups of sites that might be co-evolving using the joint distribution of substitutions [30].

- Fixed Effects Likelihood (FEL): this method locates codon sites within a gene with evidence of pervasive positive diversifying or negative selection by inferring nonsynonymous (dN) and synonymous (dS) substitution rates on a per-site basis for a given coding alignment and corresponding phylogeny [29].

- Mixed Effects Model of Evolution (MEME): a more sensitive analysis as compared to FEL, this method locates codon sites with evidence of episodic positive diversifying selection, [31].

- Relaxed Selection (RELAX): compares gene-wide selection pressure and looks for evidence that the strength of selection has been relaxed (or intensified) between the query clade and background sequences [32].

- Contrast-FEL: comparison of site-by-site selection pressure between query and background sequences to detect evidence indicative of different selective regimes [33].

- A FUBAR [34] Approach to Directional Selection (FADE): this method identifies amino-acid sites with evidence of directional selection [35]. FUBAR refers to our previously published method Fast, Unconstrained Bayesian AppRoximation for Inferring Selection.

- FitMultiModel (FMM): this method identifies genes with complex multiple instantaneous substitutions that occur within a codon, a rare but potent source of evolutionary signal. [36].

## Software availability

The RASCL application, depicted at high level in Fig 1, is implemented:

- As a standalone pipeline (https://github.com/veg/RASCL) in Snakemake [37].

- As a web application (https://galaxy.hyphy.org/u/hyphy/w/rapid-assessment-of-selection-on-clades-and-lineages), integrated as a workflow in the Galaxy [38] framework, that is freely available for use on powerful public computing infrastructure (https://usegalaxy.org).

## Visualization and downstream post-hoc analyses

Results are combined using a Python script "generate-report.py" into two machine-readable JSON files ("summary.json" and "annotation.json") that represent detailed analysis results for gene segments and individual sites, respectively. JSON is an open standard text-based file format which is also human-readable. It is well-suited for representing structured data and is commonly used for transmitting data in web applications. These JSON files can either be used as input for other software, or visualized within a standard web-browser via a feature-rich interactive RASCL dashboard hosted on ObservableHQ (Fig 2). For the web application implementation of RASCL, alignments, trees and analysis results are stored and made web-

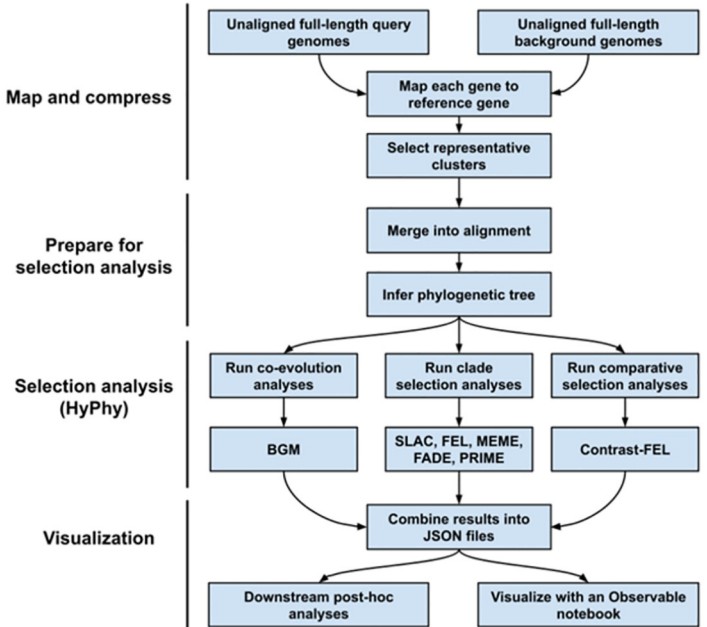

**Fig 1. The RASCL application overview.** We highlight the high-level architecture of the RASCL workflow. These include what we call multiple Phases, including a (1) Map and Compress step, where input query and background whole genome sequences are separated into individual genes from the viral genome by mapping to the reference gene. For each gene we then extract the representative gene diversity using genetic distance clustering. (2) is where we prepare our gene alignments for selection analysis. We accomplish this by merging alignments from the background and query datasets into a "combined" dataset. From this, we infer a phylogenetic tree and annotate it based on query and background sequences. (3) We perform selection analyses in HyPhy (described in further detail in the Methods section). (4) We combine the results of selection analyses across the viral genome by mapping substitutions to each position in the viral genome to create a selection 'profile' for each statistically significant site into an interpretable JSON-formatted file. These combined results are then used for further post-hoc or downstream analysis or ingested by our interactive notebook.

accessible via the Galaxy platform. Results are visualized with interactive notebooks hosted on ObservableHQ (Fig 2) that include an alignment viewer, a visualization of individual codons/ amino acid states at user-selected sites mapped onto the tips of phylogenetic trees, and detailed tabulated information on analysis results for individual genes and codon-sites.

## Results and discussion

RASCL uses molecular sequence data from genotypically distinct viral lineages to identify distinguishing features and evolution within lineages. A query set of sequences is compared against a globally diverse set of background sequences. The background data set typically contains globally circulating viral sequences, and the query data set is the set of sequences of particular interest to the user. Below, we describe our analyses of several variants of SARS-CoV-2 whole genome sequences, but RASCL is applicable to any measurably evolving pathogen with sufficient surveillance data.

### An overview of molecular surveillance of important SARS-CoV-2 viral clades

As a concrete example of the utility of RASCL consider our analysis of 112,017 BA.1 (WHO Omicron, all available BA.1 sequences as of January 2, 2022) sequences. RASCL selected a

## Sites with the highest statistical support for episodic diversifying selection in the BA.1 clade

**Table 1** List of sites found to be under diversifying positive selection by MEME
(p≤0.05) along internal branches in *BA.5*, as well as biochemical properties that are
*important* at this site (via the PRIME method)

| Coordinate (SARS-CoV-2) | Gene/ORF | Codon (in gene/ORF) | # of selected branches ▼ | p-value | q-value | Properties |
|---|---|---|---|---|---|---|
| 27381 | ORF6 | 61 | 7 | 0.0311 | 0.574 | |
| 22672 | S | 371 | 6 | 1.56e-7 | 0.000199 | |
| 22879 | S | 440 | 6 | 0.0000178 | 0.00452 | |
| 23851 | S | 764 | 6 | 0.0000179 | 0.00380 | |
| 25045 | S | 1162 | 6 | 0.0000274 | 0.00436 | |
| 22576 | S | 339 | 5 | 0.0000158 | 0.00504 | |
| 26528 | M | 3 | 4 | 0.0143 | 0.379 | |
| 26268 | E | 9 | 4 | 0.0211 | 0.497 | |
| 23074 | S | 505 | 4 | 0.0000137 | 0.00580 | |
| 25458 | ORF3a | 23 | 3 | 0.0197 | 0.481 | |
| 21640 | S | 27 | 3 | 0.00324 | 0.165 | |
| 511 | leader | 83 | 3 | 0.00153 | 0.114 | |
| 28250 | ORF8 | 120 | 3 | 0.00609 | 0.235 | |
| 22678 | S | 373 | 3 | 0.000627 | 0.0532 | |

### Partial tree of site 371 in the S gene of the BA.5 lineage

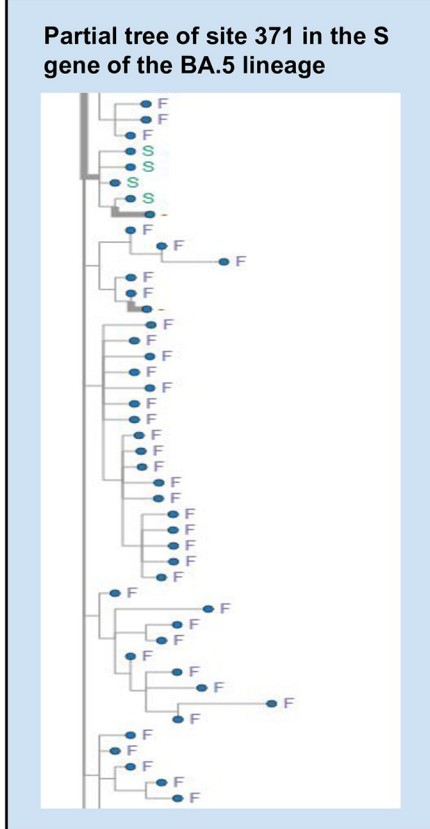

### Partial alignment of the S-gene in the BA.5 lineage

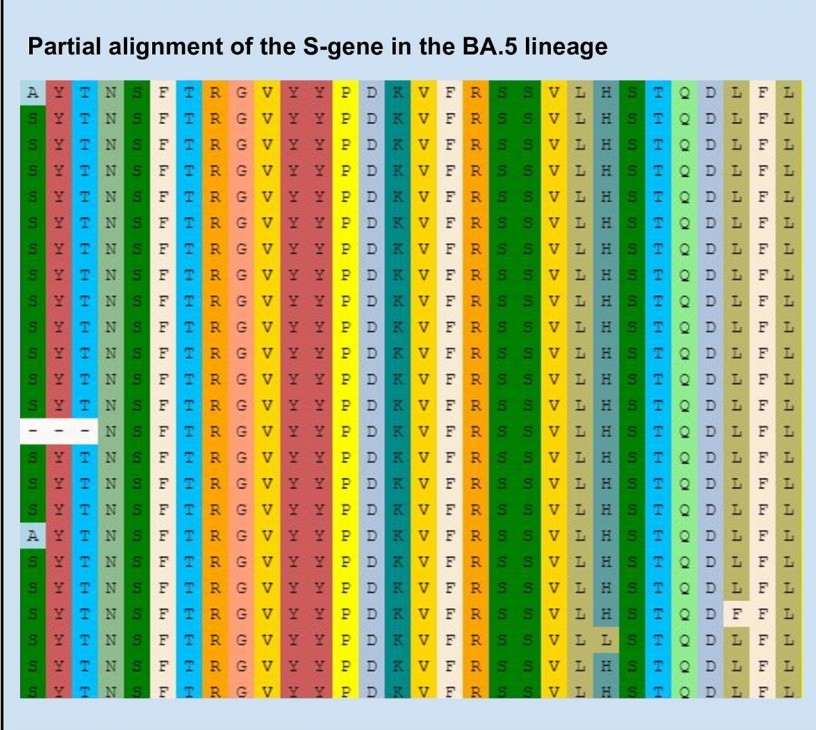

**Fig 2. Example visualization using our interactive notebook.** Here, we highlight some of the features of our interactive notebook which was created to facilitate result exploration. Key features include: (1) tables with statistically significant results for each selection analysis, (2) the ability to explore the full phylogenetic tree or a site-level tree to explore selection acting on individual sites and (3) we provide a multiple sequence alignment viewer for any of the genes in the results.

median (per gene) of 524 BA.1 sequences (a compression ratio of 99.53%) and a median of 145 background sequences (from a dataset of over 150,000 publicly available sequences from ViPR) to represent genomic diversity in SARS-CoV-2. Using the Spike gene as an exemplar, we compressed all available sequences down to 933 representative sequences, reflecting a

**Table 1. Examples of various recent RASCL analyses on variants of interest and variants of concern for SARS-CoV-2.** Full results, and more recent updates, are available through our interactive notebooks (see Methods section for additional details). The lower compression ratios for Omicron reflects the higher genomic variability of this lineage, which gave rise to many sublineages soon after emergence, combined with a high-volume of viral genome sequencing.

| Lineage | Available | Median analyzed | Compression Ratio | Positively selected sites | Positively selected sites |
|---|---|---|---|---|---|
| | | | | $p \leq 0.05$ | $p \leq 0.0001$ |
| BA.2 (WHO Omicron) | 48,623 | 319 | 1:152 | 168 | 29 |
| BA.1 (WHO Omicron) | 112,017 | 524 | 1:208 | 200 | 42 |
| B.1.617.2 (WHO Delta) | 1,983 | 59 | 1:29 | 77 | 1 |
| B.1.621 (WHO Mu) | 3,288 | 101 | 1:32 | 67 | 3 |
| C.37 (WHO Lambda) | 2,127 | 80 | 1:26 | 53 | 1 |
| P.1 (WHO Gamma) | 2,070 | 47 | 1:43 | 49 | 4 |
| B.1.1.7 (WHO Alpha) | 8,586 | 169 | 1:50 | 44 | 1 |

compression ratio of 99.17%. This level of compression is representative of recent analyses of a few VOIs/VOCs (see Table 1) with RASCL.

Importantly, there is evidence of diversifying positive selection acting on the BA.1 sequences, on 42 ($p \leq 0.0001$) individual sites (there are 4312 sites that are polymorphic in the amino-acid space among clade sequences; selection also includes the basal branch of the clade) along the internal branches of the clade (S1 Table). There is evidence of diversifying positive selection acting on the BA.1 sequences, on 359 individual sites along all branches of the clade (S2 Table), with 40 sites with an LRT p-value of $\leq 0.0001$. When comparing the strength of selection on BA.1 to background sequences along internal tree branches, 21 individual sites along the internal branches of the clade (S3 Table) showed statistically significant differences. Over the entire tree, 31 sites demonstrate evidence of directional selection (S4 Table). Along the internal branches of the BA.1 clade, 81 pairs of sites showed evidence of coevolution (S5 Table). Along the internal branches of the BA.1 clade, 47 sites showed evidence of negative selection (S6 Table). Along the internal branches of the BA.1 clade, 15 (out of 21 segments considered 71.4%) genes/ORFs showed evidence of episodic diversifying selection (BUSTED, q-value $\leq 0.1$, S7 Table).

## A closer examination of the SARS-CoV-2 BA.5 clade

We investigated the nature and extent of selective forces acting on the viral genes in BA.5 clade (all available BA.5 sequences as of August 9, 2022) sequences by performing a series of comparative phylogenetic analyses on a median of 258 BA.5 sequences and a median of 113 sequences from available sequences chosen to represent genomic diversity in SARS-CoV-2. We compiled our background dataset from the globally subsampled Nextstrain [39] (https://nextstrain.org/ncov/gisaid/global/all-time) build (last accessed June 12, 2022, genomes were sampled from the beginning of the SARS-CoV-2 pandemic) (S1 File). We observe that:

- There is evidence of diversifying positive selection acting on the BA.5 sequences, on 94 individual sites (there are 2737 sites that are polymorphic in the amino-acid space among clade sequences; selection also includes the basal branch of the clade) along the internal branches of the clade (S8 Table).

- There is evidence of diversifying positive selection acting on the BA.5 sequences, on 133 individual sites along all branches of the clade (S9 Table).

- When comparing the strength of selection on BA.5 to background sequences along internal tree branches, 12 individual sites along the internal branches of the clade (S10 Table) showed statistically significant differences.

- Over the entire tree 11 sites show evidence of directional selection (S11 Table).

- Along the internal branches of the BA.5 clade, 37 pairs of sites showed evidence of coevolution (S12 Table).

- Along the internal branches of the BA.5 clade, 14 sites showed evidence of negative selection (S13 Table).

- Along the internal branches of the BA.5 clade, 14 (out of 21 segments considered, 66.7%) genes/ORFs showed evidence of episodic diversifying selection (BUSTED[S], q-value $\leq$ 0.1, Table 2).

We consider our results from gene-wide estimates of adaptation where we observed that 3 structural and 11 non-structural proteins yield statistically significant results (Table 3). Within the set of structural proteins, we find Spike (S), Membrane glycoprotein (M), and the nucleocapsid phosphoprotein (N), these genes have been implicated in complex biological functions, including as a highly conserved target, M, [40] and have been the focus of studies on viral infection [41], pathology [42] and vaccination and therapeutic intervention [43]. Interestingly, within the set of non-structural proteins we find ORF3a, ORF6, ORF7a, and ORF8, which have been implicated in novel biological mechanisms in the SARS-CoV-2 virus including the induction of autophagy and role as a viral ion channel, ORF3a, [44, 45] disruption of nucleocytoplasmic transport, ORF6, [46] inhibition of host interferon response, ORF7a ORF8 [47, 48]. We also find

**Table 2. BUSTED[S] selection results on the BA.5 SARS-CoV-2 clade across segments.**

| Segment | omega1 | p1 | omega2 | p2 | omega3 | p3 | p | q |
|---|---|---|---|---|---|---|---|---|
| 3C | 0.03 | 0.94 | 0.07 | 0.01 | 4.93 | 0.05 | 0.2449 | 0.3429 |
| E | 0 | 0.32 | 1 | 0 | 1.18 | 0.68 | 0.4462 | 0.5511 |
| M | 0 | 0.17 | 0 | 0.79 | 8.69 | 0.04 | 0.0006 | 0.0009 |
| N | 0 | 0.93 | 0.69 | 0.05 | 51.71 | 0.02 | 0 | 0 |
| ORF3a | 0 | 0.35 | 0 | 0.61 | 25.77 | 0.04 | 0 | 0 |
| ORF6 | 0 | 0.03 | 0 | 0.93 | 34.85 | 0.04 | 0 | 0 |
| ORF7a | 0.21 | 0.91 | 0.29 | 0.05 | 28.91 | 0.03 | 0 | 0 |
| ORF8 | 0.42 | 0.96 | 0.43 | 0.01 | 45.23 | 0.02 | 0 | 0 |
| RdRp | 0.11 | 0.98 | 0.47 | 0.01 | 76.27 | 0.01 | 0 | 0 |
| S | 0.75 | 0.98 | 0.87 | 0.02 | 64042.59 | 0 | 0 | 0 |
| endornase | 0 | 0.59 | 0 | 0.39 | 18.75 | 0.03 | 0.0005 | 0.0008 |
| helicase | 0 | 0.73 | 0 | 0.09 | 2.3 | 0.17 | 0.32 | 0.4201 |
| leader | 1 | 0 | 1 | 1 | 8003.48 | 0 | 0.0002 | 0.0004 |
| methyltransferase | 0 | 0.06 | 0 | 0.93 | 67.61 | 0.01 | 0 | 0 |
| nsp10 | 0.32 | 0.49 | 0.5 | 0.51 | 1 | 0 | 0.5 | 0.5833 |
| nsp2 | 0 | 0.89 | 0.74 | 0.09 | 72.38 | 0.01 | 0 | 0 |
| nsp3 | 0 | 0.76 | 0 | 0.23 | 47.93 | 0.01 | 0 | 0 |
| nsp6 | 0 | 0.07 | 0 | 0.46 | 1 | 0.47 | 0.5 | 0.5526 |
| nsp7 | 0.32 | 0.85 | 0.34 | 0.1 | 1 | 0.05 | 0.5 | 0.525 |
| nsp8 | 0 | 0.92 | 0 | 0.07 | 34.97 | 0.02 | 0 | 0 |
| nsp9 | 0.06 | 0 | 0.36 | 1 | 1.11 | 0 | 0.5 | 0.5 |

Segment corresponds to the gene or ORF does under analysis. omega1 refers to the first omega rate class, p1 refers to proportion of sites which fit this rate class. omega2 refers to the first omega rate class, p2 refers to proportion of sites which fit this rate class. omega3 refers to the first omega rate class which captures the episodic diversifying features, p3 refers to proportion of sites which fit this rate class. p-value, the p-value for the likelihood ratio test. q-value refers to the multiple-test corrected q-value (Benjamini-Hochberg). We indicate statistically significant segments with bolded text.

**Table 3. Site profiles for selected sites in the BA.5 Spike gene.**

| # | Position | Codon | Branches | FEL | MEME | CFEL | Composition BA.5 | Composition Background |
|---|----------|-------|----------|-----|------|------|------------------|------------------------|
| 1 | 22576 | 339 | 5 | 0.00000645 | 0.0000158 | 0.000363 | D327 G15 -8 N1 | G169 D14 -1 |
| 2 | 22672 | 371 | 6 | 1.01E-07 | 1.56E-07 | 0.0075 | F310 S28 -9 L4 | S168 L8 F6 -2 |
| 3 | 22879 | 440 | 6 | 0.00000818 | 0.0000178 | 0.0716 | K298 N44 -9 | N168 K15 -1 |
| 4 | 23797 | 764 | 6 | 0.00000729 | 0.0000179 | 0.000463 | K325 N17 -8 I1 | N167 K13 -4 |
| 5 | 25045 | 1162 | 6 | 0.0000201 | 0.0000274 | 0.0208 | P308 L41 -2 | P182 Q1 S1 |

Genomic position (SARS-CoV-2): the starting coordinate of the codon in the NCBI reference SARS-CoV-2 genome. Codon in gene: the location of the codon in the corresponding Spike gene. Number Of selected branches: the number of tree branches (internal branches BA.5 clade) that have evidence of diversifying positive selection at this site (empirical Bayes factor $\geq$ 100). FEL p-value: the p-value for the likelihood ratio test that non-synonymous rate / synonymous rate $\neq$ 1 at this site. MEME Internal p-value: the p-value for the likelihood ratio test that a non-zero fraction of internal branches have omega $>$ 1 (i.e., episodic diversifying selection at this branch). CFEL p-value: the p-value for the likelihood ratio test that omega ratios between the internal branches of the two clades are different. p-values reported in this table are not corrected for multiple testing. Amino acid composition (including gaps) is reported for the BA.5 and background dataset at the corresponding site.

several members of the ORF1ab polyprotein including RNA-dependent RNA polymerase (RdRp), endoRNAse, leader, methyltransferase, nsp2, nsp3, nsp8. Several important sites in Spike from Table 1 are discussed, including: S/339, S/371, S/440, S/764, S/1162. We highlight these sites due to the level of statistical signal associated with them (we find $\geq$ 5 branches selected in the exploratory MEME analysis), and provide selection profiles for each of these sites below.

**Assessing evolutionary pressures on the SARS-CoV-2 μ (B.1.621) clade.** We identify genomic sites in B.1.621 (μ) [49] clade sequences that may be subject to selective forces and could be prioritized for further studies but have not yet reached high frequencies. We present an analysis of the B.1.621 variant, performed on individual genes/protein products, using a median of 101 μ sequences subsampled from all available sequences in GISAID [50] to represent the genomic diversity in this clade (all sequences as of September 7, 2021). A similarly subsampled global SARS-CoV-2 background dataset from publicly available sequences via the ViPR database is used as background and provided in our Github repository, (linked to in our Methods section). Interactive results for this analysis can be explored in our ObservableHQ notebook (https://observablehq.com/@aglucaci/rascl-mu) and our Virological (https://virological.org/) post (https://virological.org/t/assessing-evolutionary-pressures-on-the-sars-cov-2-mu-b-1-621-clade/760). Our analysis identified 67 (S14 Table) individual codon sites (among 1643 sites that are polymorphic in the amino-acid space) that showed evidence of episodic diversifying selection along internal branches of this clade using the MEME method at q $\leq$ 0.20 false discovery rate (FDR). A total of 5 sites (S15 Table) were found to be subject to directional selection using the FADE method.

We identify high-priority sites in SARS-CoV-2 μ (B.1.621) sequences (Table 4), with a "Rank" for each site based on a point system described below. We identify and rank sites (+1 for each category) according to the following protocol:

- Inferred to be under positive selective pressure.

- Are not clade-defining mutations.

- Contain mutations that are not predictable based on the evolution of Sarbecovirus sequences [51].

- Contain mutations that occur in a large fraction of unique haplotypes. This was shown to be predictive of near-term growth in a separate analysis from our global SARS-CoV-2 analysis [17].

**Table 4. A table of high-priority sites in SARS-CoV-2 μ B.1.621 sequences.**

| Coordinate (SARS-CoV-2) | Gene/ORF | Codon (in gene/ORF) | p-value | q-value | Rank | Property |
|---|---|---|---|---|---|---|
| 16075 | RDRP/ORF1b | 879/870Y | 0.0452115 | 0.140312 | 4 | |
| 28873 | N | 201G | 0.0460285 | 0.138086 | 4 | |
| 28253 | ORF8 | 121VHF | 0.0300385 | 0.180231 | 4 | |
| 25336 | S | 1259HV | 0.0157386 | 0.134903 | 4 | |
| 25333 | S | 1258D | 0.00852771 | 0.0959367 | 4 | |
| 13516 | RDRP/ORF1b | 26/17I | 0.0336153 | 0.168077 | 4 | |
| 14122 | RDRP/ORF1b | 228/219D | 0.036867 | 0.170155 | 4 | |
| 14530 | RDRP/ORF1b | 364/355F | 0.0416466 | 0.144161 | 4 | |
| 14767 | RDRP/ORF1b | 443/434V | 0.0384624 | 0.161005 | 4 | |
| 14785 | RDRP/ORF1b | 449/440T | 0.0383252 | 0.168257 | 4 | charge |
| 17976 | Helicase/ORF1b | 580/1503F | 0.0113156 | 0.107201 | 3 | |
| 20550 | Endornase/ORF1b | 310/2361I | 0.00645087 | 0.0893198 | 3 | |
| 21234 | Methyltransferase/ORF1b | 192/2589Y | 0.049474 | 0.134929 | 3 | |
| 21640 | S | 27LT | 0.000824192 | 0.0247258 | 3 | |
| 21997 | S | 146YNTPS | 0.000598969 | 0.0215629 | 3 | |
| 22003 | S | 148S | 0.0221244 | 0.165933 | 3 | |
| 22000 | S | 147HQNP | 0.0110123 | 0.1166 | 3 | "Overall, secondary" |
| 19482 | Exonuclease/ORF1b | 481/2005V | 0.0474395 | 0.133424 | 3 | |
| 25707 | ORF3a | 106FPIL | 0.0410847 | 0.145005 | 3 | |
| 27210 | ORF6 | 4PHI | 7.99E-06 | 0.000479141 | 3 | |
| 29023 | N | 251S | 0.0425228 | 0.141743 | 3 | |
| 19548 | Exonuclease/ORF1b | 503/2027VS | 0.0470548 | 0.134442 | 3 | |
| 29443 | N | 391AN | 0.0325738 | 0.167522 | 3 | |
| 14470 | RDRP/ORF1b | 344/335I | 0.0339005 | 0.164921 | 3 | |
| 18327 | Exonuclease/ORF1b | 96/1620I | 0.0395957 | 0.15494 | 3 | |
| 2944 | NSP3/ORF1a | 633/894S | 0.0246692 | 0.158588 | 3 | |
| 17820 | Helicase/ORF1b | 528/1451S | 0.0392635 | 0.160624 | 3 | |
| 17025 | Helicase/ORF1b | 263/1186I | 0.0241484 | 0.167182 | 3 | |
| 15001 | RDRP/ORF1b | 521/512C | 0.0457987 | 0.139725 | 3 | |
| 9472 | NSP4/ORF1a | 307/3070T | 0.040304 | 0.145094 | 3 | |
| 9424 | NSP4/ORF1a | 291/3054T | 0.0402514 | 0.147862 | 3 | |
| 9139 | NSP4/ORF1a | 196/2959S | 0.0111345 | 0.111345 | 3 | |
| 8614 | NSP4/ORF1a | 21/2784VT | 0.0400482 | 0.153376 | 2 | |
| 6535 | NSP3/ORF1a | 1273/2091D | 0.0246588 | 0.164392 | 2 | |
| 8578 | NSP4/ORF1a | 9/2772H | 0.0198022 | 0.162018 | 2 | volume |
| 19479 | Exonuclease/ORF1b | 480/2004CV | 0.00358349 | 0.0586389 | 2 | "volume, charge" |
| 8659 | NSP4/ORF1a | 36/2799N | 0.0441831 | 0.142017 | 2 | charge |
| 18960 | Exonuclease/ORF1b | 307/1831V | 0.0462054 | 0.134145 | 2 | |
| 16260 | Helicase/ORF1b | 8/931Y | 0.0309811 | 0.17989 | 2 | |
| 2230 | NSP2/ORF1a | 476/656SA | 0.0316127 | 0.172433 | 2 | |

Genomic position (in SARS-CoV-2): the starting coordinate of the codon in the NCBI reference SARS-CoV-2 genome. Gene/ORF: which gene or ORF does this site belong to. Codon in gene: the location of the codon in the corresponding gene/ORF. p-value: the p-value for the likelihood ratio test that a non-zero fraction of branches have omega > 1 (i.e., episodic diversifying selection at this branch). This is not corrected for multiple testing; the MEME test is generally conservative on real data. q-value: multiple-test corrected q-value (Benjamini-Hochberg). We assign a "Rank" to each site based on a point system described above. Properties: which, if any, of the five compositive biochemical properties [52] are conserved or changed at this site.

Additionally, where available, we provide interpretation of our identified sites in terms of known functional significance, temporal growth trends and location on the 3D structure of the protein. Briefly, each site is given a point for each of the follow requirements that are met: found to be positively selected, the site occurs outside of the clade defining site set, the site has any unexpected mutations as described in our Sarbecoviruses evolutionary analysis notebook (https://observablehq.com/@spond/sars-cov-2-pvo), the site has a mutation present above the minimum threshold in the SARS-CoV-2 Global Haplotype analysis (https://observablehq.com/@spond/sc2-haplotypes), for this we remove any mutation present in background clade, and remove gaps.

**Evidence of natural selection history operating on SARS-CoV-2 genomes.** For the set of high-priority SARS-CoV-2 genomic sites (taken from Table 4), sites inferred from B.1.621 sequences, we observe when and how selection (positively or negatively) operated on them, through a series of 3-month overlapping intervals going back to the beginning of the pandemic (Fig 3). The earliest intervals end in February 2020 and the latest—in September 2021. In selected sites we observe the temporal trends of high-priority sites in Spike (Fig 4) and RdRp (Fig 5) in B.1.621 sequences. We also describe the spatial location of sites inferred to be under positive selective pressure in the Spike gene (Fig 6) from B.1.621 (μ) sequences on the structure of the protein (from Table 4).

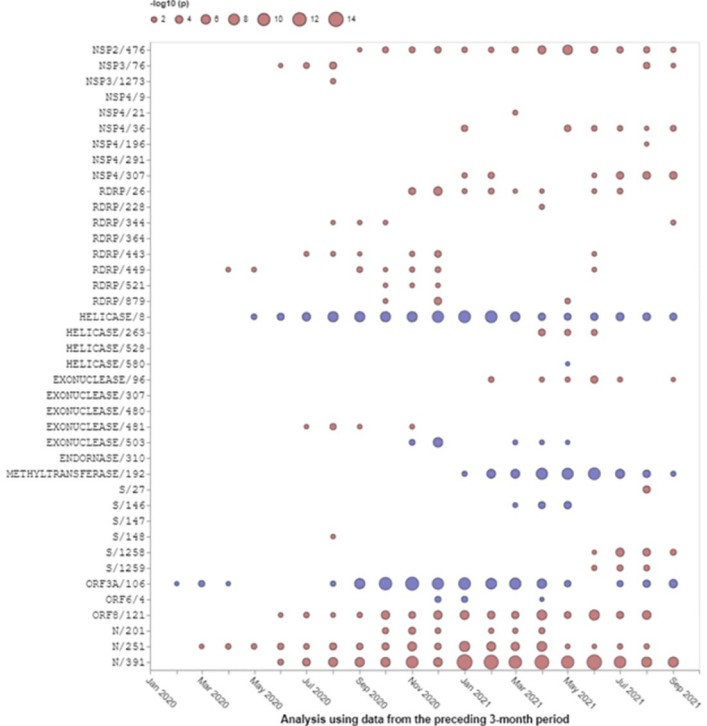

**Fig 3. Evolutionary trajectories of 40 high-priority selected sites (from Table 4).** If a site was found to be positively (red) or negatively (blue) selected during a specific time, a bubble will be drawn at a corresponding point on the plot. The area of the bubble is scaled as -log10 p, where p is the p-value of the FEL likelihood ratio test. Larger bubbles correspond to smaller p-values; p-values are not directly comparable between different time windows and different genes due to differences in sample sizes and other factors. The x-axis shows the endpoint of the time-window, e.g., March 30th, 2021, will correspond to the analysis performed with the data from January 1, 2021, to March 30, 2021. Figures like this can be generated with the "Evidence of natural selection history operating on SARS-CoV-2 genomes" ObservableHQ notebook (https://observablehq.com/@spond/sars-cov-2-selected-sites).

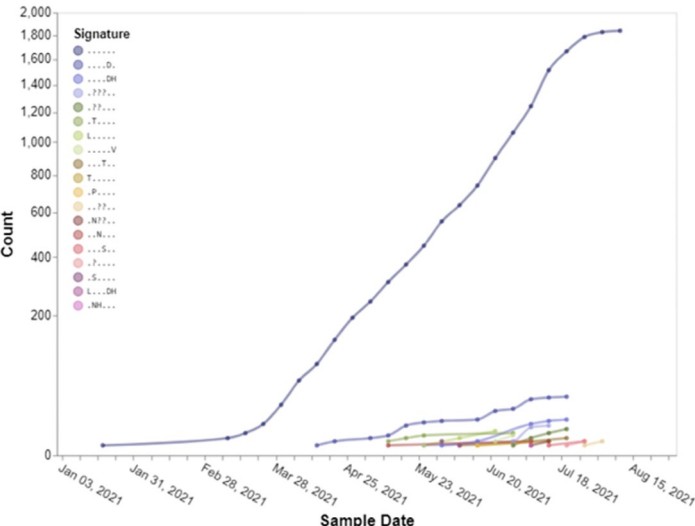

**Fig 4. Temporal trends of the substitution combinations at selected sites represented in Table 4 in the Spike gene for B.1.621 (μ) sequences in 2021 (from left to right: S/27, S/146, S/147, S/1258, S/1259).** The symbol "." denotes the reference residue at that site. Figures like this can be generated using Trends in mutational patterns across SARS-CoV-2 Spike enabled by data from https://observablehq.com/@spond/spike-trends. Additional search parameters include "B.1.621[pangolin] AND 20210101[after]". Notebook link: https://observablehq.com/@spond/spike-trends.

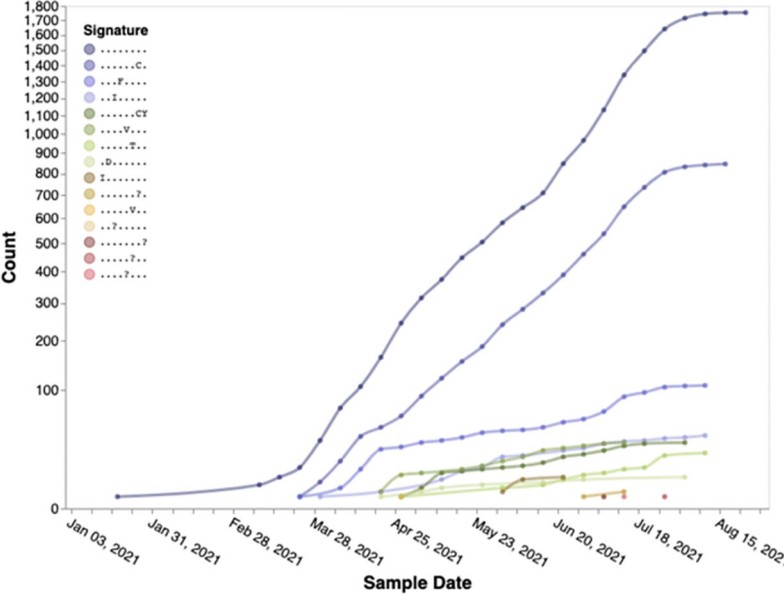

**Fig 5. Temporal trends of the substitution combinations at all sites represented in Table 4 in the RDRP (RNA-dependent RNA polymerase) gene for B.1.621 (μ) sequences in 2021 (from left to right: RDRP/26, RDRP/228, RDRP/344, RDRP/364, RDRP/443, RDRP/449, RDRP/521, RDRP/879).** The symbol "." denotes the reference residue at that site. Figures like this can be generated using Trends in mutational patterns across SARS-CoV-2 Spike enabled by data from https://observablehq.com/@spond/spike-trends. Additional search parameters include "B.1.621[pangolin] AND 20210101[after]". Notebook link https://observablehq.com/@spond/spike-trends.

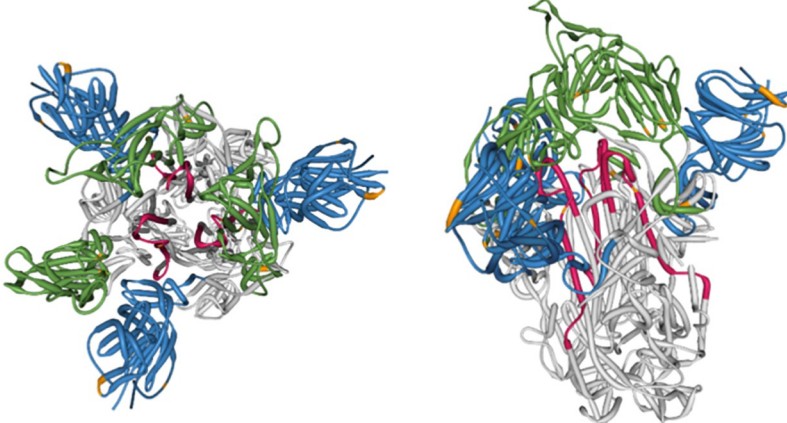

**Fig 6. Spike protein crystal structure annotation 6CRZ (https://www.rcsb.org/structure/6CRZ) with MEME sites, a measure of episodic selection (These sites are listed in Table 4).** The color legend for these figures is as follows: the N-Terminal domain (NTD) region is highlighted in Blue, the Receptor binding domain (RBD) region is highlighted in Green, The Heptad Repeat (HR) region is highlighted in Ruby, MEME (Positively selected) sites are highlighted in Orange. To interact with the figure above visit: https://observablehq.com/@aglucaci/categorical-ngl-rascl-mu.

**Potential biological and clinical significance of mutations.** The ongoing monitoring of emergent VOIs and VOCs can detect adaptive mutations before they rise to high frequency and help establish their relationship to key clinical parameters including pathogenicity and transmissibility. Additionally, continued evolution within a particular clade may form the foundation for a subclade with further functional sites of interest. Based on current information for the Spike gene from Stanford Coronavirus Antiviral Resistance Database (CoVDB, https://covdb.stanford.edu/) [53] we include several annotations with clinical relevance. From the SARS-CoV-2 B.1.621 (µ) Spike gene, we identify the following sites of interest from Table 4 due to their interaction with epitope binding in monoclonal antibodies (mAbs): 144, 145, 146, 147, 148, 417, and 501.

## Conclusions

A need that has reoccurred throughout the course of the COVID-19 pandemic is to rapidly identify molecular changes as they arise within the SARS-CoV-2 genome and to interpret how these changes impact the fitness and host-interaction of the virus. Additionally, this information is crucial to provide public health officials with the most up-to-date information when making public health decisions. To gather this information, computational and laboratory-based analytical approaches have been used to test and validate hypotheses about the observed genotype and phenotypic implications [54]. These current approaches require both significant effort and time to complete, therefore the results may be gained too slowly to inform early public health responses. Computational methods that detect natural selection can be leveraged to identify sites of interest within viral clades. Due to the massive amount of sequence availability of SARS-CoV-2 genomes, many such methods are rendered uncapable of providing results in a timely manner, bottlenecked by the increased computational complexity associated with large-scale analysis. We address this limitation with RASCL, an agile method that can be used to rapidly characterize and assess natural selection at sites and across proteins within viral genomes. Now, SARS-CoV-2 VOI/VOCs can be screened for signals of selection in a standardized manner and at an accelerated rate, while providing easily interpretable, near-real-time results.

The novelty of RASCL lies in its design; it is highly modular and easily adaptable to rapidly analyze any molecular evolving pathogen. Regarding the modularity component, there are phases of the pipeline, described in Fig 1, each of which can be parallelized across either a high-performance computing environment or a personal computer. We take the intermediate and terminal files created by our methods throughout the analysis and combine the pertinent output files together into the commonly used, standardized JSON format. To make interpretation and visualization of the results easy for the user, we created a customizable RASCL dashboard page using ObservableHQ which runs in the browser of any internet browser. The results page is dynamic and interactive, allowing the user to inspect the results for different signals of selection with ease. The modularity of RASCL makes it highly scalable, yielding near-real-time results at any stage of an outbreak. At the beginning stages of pathogen emergence when very few sequences exist, analyses run quickly, and as the outbreak persists and the number of sequences increases, subsampling can be increased, limiting the computational bottleneck.

RASCL is available in two forms, as a standalone pipeline that uses Snakemake, as well as a web application integrated as a workflow in the Galaxy framework. By implementing the method in these two ways, users at any level of bioinformatics expertise benefit, making RASCL highly accessible. RASCL has been designed such that with minimal modifications to the background and reference genomes, genes under analysis, and default thresholding settings any other evolving pathogens can be rapidly scrutinized to immediately inform public health measures. RASCL can be modified to screen for signals of selection in gene sequences from the current Monkeypox outbreak. Differences in virulence have been reported between Monkeypox isolates from two different geographic regions [55, 56], thus rapidly identifying the evolution within viral genes is highly relevant to public health efforts.

To date RASCL has been used to characterize the role of natural selection in the emergence of the Beta, Gamma, Omicron [17] and BA.4/BA.5 [57] VOC lineages, as well as to identify patterns of convergent evolution in the Alpha, Beta and Gamma lineages [11]. Whole genome sequences from any viral clade of interest (i.e., emerging pathogens), can be separated into a query and background sequence dataset representing the global diversity of viral sequences serves as the input to the tool, where the selective forces associated within and between the two sets of sequences are identified. RASCL has also been used to monitor the evolution of several lineages (see Table 1) and will be applied to future SARS-CoV-2 sublineages as they emerge. Therefore, whenever future genomic surveillance efforts reveal new potentially problematic SARS-CoV-2 lineages, we will use RASCL to analyze these too.

Among the limitations of the current study is that at this moment the RASCL application does not take recombination within genes into account; recombination between genes is handled by performing gene-by-gene analyses. While recombination plays a generally recognized role in the evolution of coronaviruses between species, only a limited amount of recombination is observed in the globally circulating viral population of SARS-CoV-2 [58]. In addition, the types and modalities of selection analyses employed in the RASCL application (described in the Methods section) have robust statistical inference that are only biased when significant recombination changes the topology of the inferred phylogenetic relationships. Future versions of RASCL will include an optional configuration to detect genetic recombination using state-of-the-art methods [59–61] and will include an updated interactive notebook to visualize and interpret these kinds of complex evolutionary signals. By taking recombination into account, we look forward to increasing the role that the RASCL application can play in the global monitoring and surveillance of evolution in SARS-CoV-2 and other important pathogens.

Our focus in this study is on the rapid, near-real-time monitoring and analysis of emerging pathogens, where the RASCL software application provides interpretable results for molecular

surveillance of continued natural evolution. While an area of active research and both proliferative and heated debate, the question of SARS-CoV-2 origins [62, 63] is out of scope for our study. Indeed, while our analysis of VOIs/VOCs in SARS-CoV-2, has uncovered important and emerging regions of interest in key viral proteins, RASCL has broader applicability to global health threats with known natural origins and existing animal reservoirs. However, further investigation in SARS-CoV-2 origins is critical for understanding the true biological and epidemiological context and complex evolutionary history [64] of global pathogens.

## Supporting information

**S1 File. GISAID accession ID's for the analyses reported in Table 1.** We also report the GISAID accession ID's for our Nextstrain background dataset.
(ZIP)

**S1 Table. List of sites found to be under diversifying positive selection by MEME ($p \leq 0.05$) along internal branches in BA.1, as well as biochemical properties that are important at this site (via the PRIME method).** Coordinate (SARS-CoV-2): the starting coordinate of the codon in the NCBI reference SARS-CoV-2 genome. Gene/ORF: which gene or ORF does this site belong to. Codon (in gene/ORF): the location of the codon in the corresponding Gene/ORF. % Of branches with omega>1: the fraction of tree branches (internal branches BA.1 clade) that have evidence of diversifying positive selection at this site (100%—pervasive selection, <100% –episodic selection). p-value: the p-value for the likelihood ratio test that a non-zero fraction of branches have omega > 1 (i.e., episodic diversifying selection at this branch). This is not corrected for multiple testing; the MEME test is generally conservative on real data. q-value: multiple-test corrected q-value (Benjamini-Hochberg). Properties: which, if any, of the five compositive biochemical properties from [52] are conserved or changed at this site.
(CSV)

**S2 Table. List of sites found to be under diversifying positive selection by MEME ($p \leq 0.05$) along all branches in BA.1, as well as biochemical properties that are important at this site (via the PRIME method).** Coordinate (SARS-CoV-2): the starting coordinate of the codon in the NCBI reference SARS-CoV-2 genome. Gene/ORF: which gene or ORF does this site belong to. Codon (in gene/ORF): the location of the codon in the corresponding Gene/ORF. % Of branches with omega>1: the fraction of tree branches (internal branches BA.1 clade) that have evidence of diversifying positive selection at this site (100%—pervasive selection, <100% –episodic selection). p-value: the p-value for the likelihood ratio test that a non-zero fraction of branches have omega > 1 (i.e., episodic diversifying selection at this branch). This is not corrected for multiple testing; the MEME test is generally conservative on real data. q-value: multiple-test corrected q-value (Benjamini-Hochberg). Properties: which, if any, of the five compositive biochemical properties from [52] are conserved or changed at this site.
(CSV)

**S3 Table. List of sites found to be selected differentially along internal branches between BA.1 and background sequences (FDR $\leq$ 0.2) using the Contrast-FEL method.** Coordinate (SARS-CoV-2): the starting coordinate of the codon in the NCBI reference SARS-CoV-2 genome. Gene/ORF: which gene or ORF does this site belong to. Codon (in gene/ORF): the location of the codon in the corresponding Gene/ORF. Ratio of omega (BA.1: reference): the ratio of site-level omega estimates for the two sets of branches. If this ratio is > 1, then selection on BA.1 is stronger. These values are highly imprecise and should be viewed as qualitative measures. p-value: the p-value for the likelihood ratio test that omega ratios between the

internal branches of the two clades are different. This is not corrected for multiple testing. q-value: multiple-test corrected q-value (Benjamini-Hochberg).
(CSV)

**S4 Table. List of sites found to be evolving under directional selection in the entire tree, using a FUBAR-like implementation of the DEPS [35] method.** The BA.1 tree was rooted on the genome reference sequence for this analysis. Coordinate (SARS-CoV-2): the starting coordinate of the codon in the NCBI reference SARS-CoV-2 genome Gene/ORF: which gene or ORF does this site belong to. Codon (in gene/ORF): the location of the codon in the corresponding Gene/ORF. Target amino-acid: which amino-acids have statistical support (Bayes Factor $\geq$ 100) for accelerated evolution towards them.
(CSV)

**S5 Table. Pairs of sites (BA.1) found to have epistatic (co-evolving) substitution patterns by BGM method.** Gene/ORF: which gene or ORF does this site belong to. Codon 1/2 (in gene/ORF): the location of the two interacting codons in the corresponding Gene/ORF. Posterior probability of non-independence: estimated posterior probability that substitutions which occur on the interior branches of the BA.1 clade are not independent.
(CSV)

**S6 Table. List of sites found to be under pervasive negative selection by FEL (p$\leq$0.05) along internal branches in BA.1.** Coordinate (SARS-CoV-2): the starting coordinate of the codon in the NCBI reference SARS-CoV-2 genome. Gene/ORF: which gene or ORF does this site belong to. Codon (in gene/ORF): the location of the codon in the corresponding Gene/ORF. Synonymous rate: Site estimate for the synonymous substitution rate (alpha). These values are highly imprecise and should be viewed as qualitative measures. Non-synonymous rate: Site estimate for the non-synonymous substitution rate (beta). These values are highly imprecise and should be viewed as qualitative measures. p-value: the p-value for the likelihood ratio test that beta / alpha $\neq$ 1. q-value: multiple-test corrected q-value (Benjamini-Hochberg).
(CSV)

**S7 Table. BUSTED[S] selection results on the BA.1 SARS-CoV-2 clade across segments.** Segment corresponds to the gene or ORF does under analysis. Omega1 refers to the first omega rate class, p1 refers to proportion of sites which fit this rate class. Omega2 refers to the first omega rate class, p2 refers to proportion of sites which fit this rate class. Omega3 refers to the first omega rate class which captures the episodic diversifying features, p3 refers to proportion of sites which fit this rate class. P-value, the p-value for the likelihood ratio test. Q-value refers to the multiple-test corrected q-value (Benjamini-Hochberg).
(CSV)

**S8 Table. List of sites found to be under diversifying positive selection by MEME (p$\leq$0.05) along internal branches in BA.5, as well as biochemical properties that are important at this site (via the PRIME method).** Coordinate (SARS-CoV-2): the starting coordinate of the codon in the NCBI reference SARS-CoV-2 genome. Gene/ORF: which gene or ORF does this site belong to. Codon (in gene/ORF): the location of the codon in the corresponding Gene/ORF. % Of branches with omega>1: the fraction of tree branches (internal branches BA.1 clade) that have evidence of diversifying positive selection at this site (100%—pervasive selection, <100% −episodic selection). p-value: the p-value for the likelihood ratio test that a non-zero fraction of branches have omega > 1 (i.e., episodic diversifying selection at this branch). This is not corrected for multiple testing; the MEME test is generally conservative on real data.

q-value: multiple-test corrected q-value (Benjamini-Hochberg). Properties: which, if any, of the five compositive biochemical properties from [52] are conserved or changed at this site. (CSV)

**S9 Table. List of sites found to be under diversifying positive selection by MEME (p≤0.05) along all branches in BA.5, as well as biochemical properties that are important at this site (via the PRIME method).** Coordinate (SARS-CoV-2): the starting coordinate of the codon in the NCBI reference SARS-CoV-2 genome. Gene/ORF: which gene or ORF does this site belong to. Codon (in gene/ORF): the location of the codon in the corresponding Gene/ORF. % Of branches with omega>1: the fraction of tree branches (internal branches BA.1 clade) that have evidence of diversifying positive selection at this site (100%—pervasive selection, <100% –episodic selection). p-value: the p-value for the likelihood ratio test that a non-zero fraction of branches have omega > 1 (i.e., episodic diversifying selection at this branch). This is not corrected for multiple testing; the MEME test is generally conservative on real data. q-value: multiple-test corrected q-value (Benjamini-Hochberg). Properties: which, if any, of the five compositive biochemical properties from [52] are conserved or changed at this site. (CSV)

**S10 Table. List of sites found to be selected differentially along internal branches between BA.5 and background sequences (FDR ≤ 0.2) using the Contrast-FEL method.** Coordinate (SARS-CoV-2): the starting coordinate of the codon in the NCBI reference SARS-CoV-2 genome. Gene/ORF: which gene or ORF does this site belong to. Codon (in gene/ORF): the location of the codon in the corresponding Gene/ORF. Ratio of omega (BA.1: background): the ratio of site-level omega estimates for the two sets of branches. If this ratio is > 1, then selection on BA.1 is stronger. These values are highly imprecise and should be viewed as qualitative measures. p-value: the p-value for the likelihood ratio test that omega ratios between the internal branches of the two clades are different. This is not corrected for multiple testing. q-value: multiple-test corrected q-value (Benjamini-Hochberg). (CSV)

**S11 Table. List of sites found to be evolving under directional selection in the entire tree, using a FUBAR-like implementation of the DEPS method.** The BA.5 tree was rooted on the genome reference sequence for this analysis. Coordinate (SARS-CoV-2): the starting coordinate of the codon in the NCBI reference SARS-CoV-2 genome Gene/ORF: which gene or ORF does this site belong to. Codon (in gene/ORF): the location of the codon in the corresponding Gene/ORF. Target amino-acid: which amino-acids have statistical support (Bayes Factor ≥ 100) for accelerated evolution towards them. (CSV)

**S12 Table. Pairs of sites (BA.5) found to have epistatic (co-evolving) substitution patterns by BGM method.** Gene/ORF: which gene or ORF does this site belong to. Codon 1/2 (in gene/ORF): the location of the two interacting codons in the corresponding Gene/ORF. Posterior probability of non-independence: estimated posterior probability that substitutions which occur on the interior branches of the BA.1 clade are not independent. (CSV)

**S13 Table. List of sites found to be under pervasive negative selection by FEL (p≤0.05) along internal branches in BA.5.** Coordinate (SARS-CoV-2): the starting coordinate of the codon in the NCBI reference SARS-CoV-2 genome. Gene/ORF: which gene or ORF does this site belong to. Codon (in gene/ORF): the location of the codon in the corresponding Gene/ORF. Synonymous rate: Site estimate for the synonymous substitution rate (alpha). These

values are highly imprecise and should be viewed as qualitative measures. Non-synonymous rate: Site estimate for the non-synonymous substitution rate (beta). These values are highly imprecise and should be viewed as qualitative measures. p-value: the p-value for the likelihood ratio test. "q-value": multiple-test corrected q-value (Benjamini-Hochberg).
(CSV)

**S14 Table. List of sites found to be under diversifying positive selection by MEME (p≤0.05) along internal branches in B.1.621, as well as biochemical properties that are important at this site (via the PRIME method).** Coordinate (SARS-CoV-2): the starting coordinate of the codon in the NCBI reference SARS-CoV-2 genome. Gene/ORF: which gene or ORF does this site belong to. Codon (in gene/ORF): the location of the codon in the corresponding Gene/ORF. % of branches with omega > 1: the fraction of tree branches (internal branches BA.1 clade) that have evidence of diversifying positive selection at this site (100%—pervasive selection, <100% –episodic selection). p-value: the p-value for the likelihood ratio test that a non-zero fraction of branches have omega > 1 (i.e., episodic diversifying selection at this branch). This is not corrected for multiple testing; the MEME test is generally conservative on real data. q-value: multiple-test corrected q-value (Benjamini-Hochberg). Properties: which, if any, of the five compositive biochemical properties from [52] are conserved or changed at this site.
(CSV)

**S15 Table. List of sites found to be evolving under directional selection in the entire tree, using a FUBAR-like implementation of the DEPS method.** The B.1.621 tree was rooted on the genome reference sequence for this analysis. Coordinate (SARS-CoV-2): the starting coordinate of the codon in the NCBI reference SARS-CoV-2 genome Gene/ORF: which gene or ORF does this site belong to. Codon (in gene/ORF): the location of the codon in the corresponding Gene/ORF. Target amino-acid: which amino-acids have statistical support (Bayes Factor ≥ 100) for accelerated evolution towards them.
(CSV)

## Acknowledgments

We gratefully acknowledge all data contributors, i.e., the Authors and their Originating laboratories responsible for obtaining the specimens, and their Submitting laboratories for generating the genetic sequence and metadata and sharing via the GISAID Initiative, on which this research is based. We thank the global community of health-care workers and scientists who work tirelessly to face the pandemic head-on. We thank members of the Datamonkey and HyPhy, and Galaxy teams for their continued assistance in the development and application of our software.

## Author Contributions

**Conceptualization:** Alexander G. Lucaci, Sergei L. Kosakovsky Pond.

**Data curation:** Alexander G. Lucaci, Stephen D. Shank.

**Formal analysis:** Alexander G. Lucaci, Anton Nekrutenko, Darren P. Martin, Sergei L. Kosakovsky Pond.

**Funding acquisition:** Sergei L. Kosakovsky Pond.

**Investigation:** Alexander G. Lucaci, Han Mei, Anton Nekrutenko, Darren P. Martin, Sergei L. Kosakovsky Pond.

**Methodology:** Alexander G. Lucaci, Jordan D. Zehr, Alexander Ostrovsky, Han Mei, Anton Nekrutenko, Darren P. Martin, Sergei L. Kosakovsky Pond.

**Project administration:** Alexander G. Lucaci, Darren P. Martin, Sergei L. Kosakovsky Pond.

**Resources:** Alexander G. Lucaci, Stephen D. Shank, Dave Bouvier, Anton Nekrutenko, Darren P. Martin, Sergei L. Kosakovsky Pond.

**Software:** Alexander G. Lucaci, Jordan D. Zehr, Stephen D. Shank, Dave Bouvier, Alexander Ostrovsky, Anton Nekrutenko, Sergei L. Kosakovsky Pond.

**Supervision:** Alexander G. Lucaci, Anton Nekrutenko, Darren P. Martin, Sergei L. Kosakovsky Pond.

**Validation:** Alexander G. Lucaci, Jordan D. Zehr, Anton Nekrutenko, Sergei L. Kosakovsky Pond.

**Visualization:** Alexander G. Lucaci, Jordan D. Zehr, Stephen D. Shank, Sergei L. Kosakovsky Pond.

**Writing – original draft:** Alexander G. Lucaci, Anton Nekrutenko, Darren P. Martin, Sergei L. Kosakovsky Pond.

**Writing – review & editing:** Alexander G. Lucaci, Darren P. Martin, Sergei L. Kosakovsky Pond.

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
