## [Decision Letter · Decision Letter 0]

20 Jul 2022

PONE-D-22-06403

RASCL: RAPID ASSESSMENT OF SELECTION IN CLADES THROUGH MOLECULAR SEQUENCE ANALYSIS

PLOS ONE

Dear Dr. Lucaci,

Thank you for submitting your manuscript to PLOS ONE; I sincerely apologise for the unusually delayed review timeframe.

Your manuscript has been assessed by one reviewer, whose comments are appended below. After careful consideration, we feel that it has merit but does not fully meet PLOS ONE’s publication criteria as it currently stands. Therefore, we invite you to submit a revised version of the manuscript that addresses the points raised during the review process.

Among the limitations raised by the reviewer is that the documentation for the software is incomplete; please ensure that the software meets PLOS ONE's policies for sharing software (https://journals.plos.org/plosone/s/materials-software-and-code-sharing#loc-sharing-software).

Please note that we have only been able to secure a single reviewer to assess your manuscript. We are issuing a decision on your manuscript at this point to prevent further delays in the evaluation of your manuscript. Please be aware that the editor who handles your revised manuscript might find it necessary to invite additional reviewers to assess this work once the revised manuscript is submitted. However, we will aim to proceed on the basis of this single review if possible.

We look forward to receiving your revised manuscript.

Kind regards,

Emily Chenette

Editor in Chief

PLOS ONE

“DPM is funded by the Wellcome Trust (222574/Z/21/Z). This research was supported in part by grants R01 AI134384 (NIH/NIAID) and grant 2027196 (NSF/DBI,BIO) to AN and SLKP. The funding bodies played no role in the design of the study, the collection, analysis, and interpretation of data, nor in writing the manuscript.”

We note that you have provided funding information that is not currently declared in your Funding Statement. However, funding information should not appear in the Funding section or other areas of your manuscript. We will only publish funding information present in the Funding Statement section of the online submission form.

“DPM is funded by the Wellcome Trust (222574/Z/21/Z). This research was supported in

part by grants R01 AI134384 (NIH/NIAID) and grant 2027196 (NSF/DBI,BIO) to AN and SLKP.

The funding bodies played no role in the design of the study, the collection, analysis, and

interpretation of data, nor in writing the manuscript.”

“None”

Reviewers' comments:

Reviewer's Responses to Questions

**Comments to the Author**

1. Is the manuscript technically sound, and do the data support the conclusions?

Reviewer #1: Yes

2. Has the statistical analysis been performed appropriately and rigorously? 

Reviewer #1: Yes

3. Have the authors made all data underlying the findings in their manuscript fully available?

Reviewer #1: No

4. Is the manuscript presented in an intelligible fashion and written in standard English?

Reviewer #1: Yes

5. Review Comments to the Author

Reviewer #1: This manuscript is structured like an "application note", although I am not aware of this journal having this type of article. Thus, the text is very brief and many aspects of the analysis are not explained to the level that I would expect for a standard manuscript. However, the methods presented in the manuscript are important and timely, representing significant advances in our ability to detect selection in large amounts of SARS-CoV-2 genome data.

Please clarify the role of the query/background partition of sequence data in the analysis. What are the respective roles of the different molecular evolution models employed in this workflow, e.g., SLAC, BGM, FEL, etc.? Since many of these are various tests for selection, how are their outputs integrated into a meaningful whole? You should clarify the distinction between "pervasive" and "episodic" selection. The paper would be more accessible to a broader audience if it provided at least a brief explanation of what each tool does.

I found this software unnecessarily difficult to run. It requires some specific computing environments (Anaconda and Slurm) and the documentation is very incomplete. It should be possible to run RASCL via the Snakemake file on a single Linux workstation with a sufficient number of cores and RAM. The package does not appear to include sufficient example files for running a demo (there is `TEST.fasta`, but no `BA.5-04132022`). The Galaxy web service for this tool was not available.

Please provide line numbers in the revised manuscript. If you are using LaTeX, please use the url package to allow line breaks at slashes in long URLs.

Specific comments:

* Abstract, "sensitivity to sequencing errors" and also first page "because noisy sequencing data" -- I think the compressed time-scale and depth of sampling of the SARS-CoV-2 pandemic are important factors in the difficulty of deploying standard selection analyses. More specifically, they are the reason why sequencing error is more problematic than usual.

* "Since epidemiologically relevant mutations [...]", can you please clarify what this means? For example, mutations that affect transmission rates.

* Figure 1, the flowchart is very difficult to read with a poor text to whitespace ratio, and the figure legend is not very helpful. What are "clade genomes"?

* "It is not necessary to remove sequences in the query dataset that are duplicated in the reference dataset -- the pipeline will do this automatically." How does this step handle sequences with ambiguous (uncalled) bases? How does it decide which duplicate sequence to retain as the representative sequence?

* Please explain what ViPR is.

* reversed left square bracket by "RAxML-NG, Kozlov"

* "To mitigate the potentially confounding influences of within-host evolution [...]" How would within-host evolution confound our ability to detect among-host evolution in response to selection?

* JSON (JavaScript Object Notation) should be defined at first usage. It would also be helpful to explain that it is a standard, human-readable format for the interchange of serialized data online, or something to that effect.

* Table 1, what is the background clade for these analyses?

* "Of note, the lower compression ratios for Omicron may reflect its rapid detection, and a mature sequencing ecosystem." Please expand on this. What do you mean by a "mature sequencing ecosystem" and how would this affect the compression ratio?

* Availability and requirements: the HyPhy batch language is highly application-specific that will be unfamiliar to the broader research community, and the language itself has undergone extensive revisions in recent years. Is there an up-to-date language specification available for new users?

* The web service https://usegalaxy.eu/u/hyphy/w/rascl was unavailable when I attempted to access it for this review (HTTP error 500). However, the main page https://usegalaxy.eu/ was working fine. RASCL was not listed in available tools, and I could not locate it using the search interface.

* The installation instructions in GitHub `README` are missing a `cd RASCL` step between 1 and 2. Also, the last `RASCL` argument in the `git clone` command is not strictly necessary, since it is the default value.

* I got the following error when following the installation instructions:

```

ResolvePackageNotFound:

- stephenshank::tn93=1.0.7

```

I'm running conda version 4.11.0 on macOS. I didn't run into this problem on my Linux workstation, so I assume that the issue is that there is no macOS binary distributed via this package - however, there is one available via `bioconda::tn93`. Is there any reason why the RASCL environment can't point to that package?

* There is no file named `snakemake_config.json` in the RASCL directory. The `README.md` configuration instructions should be corrected to refer to `config.json`.

* The configuration instructions are unclear. If the "clade of interest" is set to `B.1.1.7`, does that refer to a FASTA filename (user input data), or to labels contained within that FASTA file? Why don't these instructions employ the same terminology as the manuscript, *i.e.*, "query" and "background"? In the file `cluster.json`, what do the labels `cluster`, `nodes`, `ppn` and `name` correspond to?

* Running RASCL appears to require the Slurm workload manager (`qsub`). If so, then this should be listed as a package dependency. However, users should have the option of running RASCL without a workload manager.

* observablehq.com notebook has older GitHub link, https://github.com/veg/SARS-CoV-2_Clades

* "Label tree with amino-acids" checkbox does not seem to be working.

* My JS console is listing a lot of errors on page load that appear to be associated with phylotree.js; for example:

* Error: <path> attribute d: Expected number, "MNaN,0LNaN,NaNLNa…".

* Error: <g> attribute transform: Expected number, "translate (NaN,NaN) ".

* for the notebook interface, is it possible to provide some visual cue to indicate that table rows can be sorted by clicking on column labels?

* can the developers please provide an additional set of instructions for running RASCL in a Linux environment that is not running Anaconda? Some shared computing environments prohibit users from using Anaconda, e.g., Compute Canada.

* please provide instructions for retrieving JSON data from the observablehq site via some API.

* The Python scripts have some weird formatting. For example, in `tn93_cluster.py` there are long spaces within `add_argument()` calls, each on a single line. Please conform to PEP 8 conventions if possible. `generate-report.py` has a monolithic for-loop spanning over 500 lines. This code would be much more maintainable if the developers applied a more modular code style. Also see lines 689-695 and 743-749 for weird whitespace.</g></path>

6. PLOS authors have the option to publish the peer review history of their article (what does this mean?). If published, this will include your full peer review and any attached files.

Reviewer #1: **Yes: **Art Poon

---

## [Author Response · Author response to Decision Letter 0]

30 Aug 2022

Aug 26, 2022

Dr. Emily Chenette

Editor in Chief

PLOS ONE

Dear Dr. Chenette,

RE: Response to reviewers for: RASCL: RAPID ASSESSMENT OF SELECTION IN CLADES THROUGH MOLECULAR SEQUENCE ANALYSIS.

Thank you for the opportunity to submit a revised manuscript on “RASCL: RAPID ASSESSMENT OF SELECTION IN CLADES THROUGH MOLECULAR SEQUENCE ANALYSIS”. Please find attached our revised contribution that incorporates responses to the Editor and Reviewer comments. Each of the comments have been addressed and a detailed response is attached. Both a marked-up manuscript and a clean LaTeX version of the paper are included.

The authors have declared that no competing interests exist.

The amended Funding Statement is as follows: DPM is funded by the Wellcome Trust (222574/Z/21/Z). This research was supported in part by grants R01 AI134384 (NIH/NIAID) and grant 2027196 (NSF/DBI,BIO) to AN and SLKP. The funding bodies played no role in the design of the study, the collection, analysis, and interpretation of data, nor in writing the manuscript. 

Yours sincerely,

Alexander G. Lucaci, M.S.

Ph.D. Candidate

Institute for Genomics and Evolutionary Medicine (iGEM)

Temple University

Editor comments

Comment #1 

Please ensure that your manuscript meets PLOS ONE's style requirements, including those for file naming. The PLOS ONE style templates can be found at https://journals.plos.org/plosone/s/file?id=wjVg/PLOSOne_formatting_sample_main_body.pdf and https://journals.plos.org/plosone/s/file?id=ba62/PLOSOne_formatting_sample_title_authors_affiliations.pdf.

Response 1: We have modified our manuscript to meet the PLOS ONE style requirements, including those for file naming. These changes are reflected in the revised manuscript.

Comment #2

Please update your submission to use the PLOS LaTeX template. The template and more information on our requirements for LaTeX submissions can be found at http://journals.plos.org/plosone/s/latex.

Response 2: We have updated our revised manuscript to use the PLOS LaTex template.

Comment #3

Thank you for stating the following in the Funding Section of your manuscript:

“DPM is funded by the Wellcome Trust (222574/Z/21/Z). This research was supported in part by grants R01 AI134384 (NIH/NIAID) and grant 2027196 (NSF/DBI,BIO) to AN and SLKP. The funding bodies played no role in the design of the study, the collection, analysis, and interpretation of data, nor in writing the manuscript.”

We note that you have provided funding information that is not currently declared in your Funding Statement. However, funding information should not appear in the Funding section or other areas of your manuscript. We will only publish funding information present in the Funding Statement section of the online submission form.

“DPM is funded by the Wellcome Trust (222574/Z/21/Z). This research was supported in

part by grants R01 AI134384 (NIH/NIAID) and grant 2027196 (NSF/DBI,BIO) to AN and SLKP.

The funding bodies played no role in the design of the study, the collection, analysis, and

interpretation of data, nor in writing the manuscript.” Please include your amended statements within your cover letter; we will change the online submission form on your behalf.

Response 3: The requested changes are reflected in our revised manuscript, we have removed the Funding statement from the revised manuscript. We have included the amended Funding Statement section in our cover letter.

Comment #4

Thank you for stating the following in your Competing Interests section: 

“None”

Response 4: We have included the amended Competing Interests section in our cover letter.

Reviewer comments

Comment #1

Have the authors made all data underlying the findings in their manuscript fully available? The PLOS Data policy requires authors to make all data underlying the findings described in their manuscript fully available without restriction, with rare exception (please refer to the Data Availability Statement in the manuscript PDF file). The data should be provided as part of the manuscript or its supporting information, or deposited to a public repository. For example, in addition to summary statistics, the data points behind means, medians and variance measures should be available. If there are restrictions on publicly sharing data—e.g. participant privacy or use of data from a third party—those must be specified. Reviewer #1: No

Response 1: Our ability to make all underlying data publicly available is restricted under the GISAID Database Access Agreement, which states that “You will not distribute, redistribute, share, or otherwise make available Data, to any third party or the public, unless the individual is an Authorized User of GISAID”. However, for authorized users, we are able to provide a File (S1 File of the revised manuscript) of sequence identifiers for data retrieval and results replication. 

Comment #2

This manuscript is structured like an "application note", although I am not aware of this journal having this type of article. Thus, the text is very brief and many aspects of the analysis are not explained to the level that I would expect for a standard manuscript.

Response 2: We have modified our manuscript to provide additional details, without unnecessary bloat

Comment #3

Please clarify the role of the query/background partition of sequence data in the analysis.

Response 3: We have included a clarification of the respective roles of query and background sequence data.

Comment #4

What are the respective roles of the different molecular evolution models employed in this workflow, e.g., SLAC, BGM, FEL, etc.? Since many of these are various tests for selection, how are their outputs integrated into a meaningful whole? You should clarify the distinction between "pervasive" and "episodic" selection.

Response 4: Each method is designed to ask and answer specific biological and statistical questions. We have modified the manuscript to further explain their relative roles. 

Comment #5

I found this software unnecessarily difficult to run. It requires some specific computing environments (Anaconda and Slurm) and the documentation is very incomplete. It should be possible to run RASCL via the Snakemake file on a single Linux workstation with a sufficient number of cores and RAM. The package does not appear to include sufficient example files for running a demo (there is `TEST.fasta`, but no `BA.5-04132022`). The Galaxy web service for this tool was not available.

Response 5: We have expanded our README file to be more user friendly. This includes information about simple demo runs, advanced configuration, and how to run locally.

We have provided a “run_Local.sh” version of the submitting script to facilitate the user running the program on a local machine. We have also updated the demo data, reflected in the “/data/Example1” folder within our repository, to be the default configuration for the demo run.

We have updated the Galaxy web service link to reflect a stable link at https://galaxy.hyphy.org/u/hyphy/w/rapid-assessment-of-selection-on-clades-and-lineages. Should the link become unreachable in the future, we also provide a backup link to our workflow via our dedicated Github repository. 

Comment #6

Please provide line numbers in the revised manuscript. If you are using LaTeX, please use the url package to allow line breaks at slashes in long URLs.

Response 6: We have added line numbers to our revised manuscript.

Comment #7

* Abstract, "sensitivity to sequencing errors" and also first page "because noisy sequencing data" -- I think the compressed time-scale and depth of sampling of the SARS-CoV-2 pandemic are important factors in the difficulty of deploying standard selection analyses. More specifically, they are the reason why sequencing error is more problematic than usual.

Response 7: A compressed time-scale and depth of sampling can lead to a high number of duplicates, where not enough time has passed to observe inter-host transmission variability of the virus. Sequencing errors are also due in large part to the global scientific community figuring out a way to best process SC2 samples, and to develop standards in protocols, assembly, which also contribute to noise. We address these partially by using Internal branches on some of our analyses, to reduce noise (we explain this further below). However, the scientific space and global community has matured since early in the days of the pandemic. What we face now is still the contribution of some noise, but highly dense (i.e. low-temporal sampling) leading to a significant number of duplicate and near-duplicate sequences.

Comment #8

* "Since epidemiologically relevant mutations [...]", can you please clarify what this means? For example, mutations that affect transmission rates.

Response 8: Here, we refer to epidemiological mutations in spike that received early attention such as D614G as well those that followed the emergence of seed variants. 

Comment #9

* Figure 1, the flowchart is very difficult to read with a poor text to whitespace ratio, and the figure legend is not very helpful. What are "clade genomes"?

Response 9: Figure 1 has been expanded into Figures 1 and 2 to add clarity. Additionally, we have modified the figure legends text to provide a stronger description of the figures.

Comment #10

* "It is not necessary to remove sequences in the query dataset that are duplicated in the reference dataset -- the pipeline will do this automatically." How does this step handle sequences with ambiguous (uncalled) bases? How does it decide which duplicate sequence to retain as the representative sequence?

Response 10: This is done using the standard TN93 distance calculation (tn93-cluster), where the default is to RESOLVE ambiguities (e.g. R will “match” A or G, N will “match” any resolved based). Among all the sequences that are placed in the “duplicate” bin, the one that has the fewest overall ambiguities (fraction of sequence length) will be retained. Because most of the ambiguities in SARS-CoV-2 consensus genomes are ‘N’, this has the effect of selecting the “least ambiguous” sequence. In addition to this, we further “mask” (with ‘---’, using strike_ambigs.bf script) partially resolved codons (e.g. ANC) in “post-compression” MSA prior to submitting them to HyPhy. This is because PARTIALLY resolved codons with missing data (N) can create false positive selection signals along very short tree branches. 

Comment #11

* Please explain what ViPR is.

Response 11: We have modified the manuscript to include a description for ViPR.

Comment #12

* reversed left square bracket by "RAxML-NG, Kozlov"

Response 12: Fixed.

Comment #13

* "To mitigate the potentially confounding influences of within-host evolution [...]" How would within-host evolution confound our ability to detect among-host evolution in response to selection?

Response 13: This comment refers to the observation that many within-host mutations are maladaptive at the population level (e.g. https://www.ncbi.nlm.nih.gov/pmc/articles/PMC1480537/ and https://academic.oup.com/mbe/article/24/3/845/1246056). We added a clarification to the text, as follows: 

“To partially mitigate the potentially confounding influences of within-host evolution, where mutations occurring within an individual have not been filtered by selection at the broader population-level, and sequencing errors, these analyses are performed only on the internal branches of phylogenetic trees, where at least one or more rounds of virus transmission are captured…”

Comment #14

* JSON (JavaScript Object Notation) should be defined at first usage. It would also be helpful to explain that it is a standard, human-readable format for the interchange of serialized data online, or something to that effect.

Response 14: We have modified the text as requested.

Comment #15

* Table 1, what is the background clade for these analyses?

Response 15: The background clade for these analyses is a curated dataset from ViPR, it is also publically available on our dedicated Github repository at: https://github.com/veg/RASCL/tree/main/data/ReferenceSetViPR

Comment #16

* "Of note, the lower compression ratios for Omicron may reflect its rapid detection, and a mature sequencing ecosystem." Please expand on this. What do you mean by a "mature sequencing ecosystem" and how would this affect the compression ratio?

Response 16: The Omicron variants designation as a VOC on November 26, 2021 occurred in a significantly more mature scientific ecosystem than that of earlier variants. Many Omicron variant sequences were made available on public databases such as GISAID through the concerted effort of experienced public health laboratories and their respective teams. The rapid pace of sequencing through the ability to better detect, sample, sequence, and assemble the SARS-CoV-2 viral genome, which was not previously available at the beginning of the pandemic.

Comment #17

* Availability and requirements: the HyPhy batch language is highly application-specific that will be unfamiliar to the broader research community, and the language itself has undergone extensive revisions in recent years. Is there an up-to-date language specification available for new users?

Response 17: We have designed the RASCL application to run with minimal user configuration. We therefore expect very little programming background, especially with our user-friendly Galaxy workflow, which provides a point and click interface. The end user does not need to tinker with the underlying HyPhy batch language, methods, or application in order to successfully complete the analysis.

Comment #18

* The web service https://usegalaxy.eu/u/hyphy/w/rascl was unavailable when I attempted to access it for this review (HTTP error 500). However, the main page https://usegalaxy.eu/ was working fine. RASCL was not listed in available tools, and I could not locate it using the search interface.

Response 18: We have updated the Galaxy web service link to reflect a stable link at https://galaxy.hyphy.org/u/hyphy/w/rapid-assessment-of-selection-on-clades-and-lineages. Should the link become unreachable in the future, we also provide a backup link to our workflow via our dedicated Github repository. 

Comment #19

* The installation instructions in GitHub `README` are missing a `cd RASCL` step between 1 and 2. Also, the last `RASCL` argument in the `git clone` command is not strictly necessary, since it is the default value.

Response 19: We have modified the README file to reflect these changes.

Comment #20

* I got the following error when following the installation instructions:

```

ResolvePackageNotFound:

- stephenshank::tn93=1.0.7

```

I'm running conda version 4.11.0 on macOS. I didn't run into this problem on my Linux workstation, so I assume that the issue is that there is no macOS binary distributed via this package - however, there is one available via `bioconda::tn93`. Is there any reason why the RASCL environment can't point to that package?

Response 20: The “bioconda::tn93" package was an outdated version that did not provide key functionality required for the RASCL application. We successfully reached out to the bioconda team to provide the latest available version, and we now point to it in our environment file. Additionally, we have significantly improved usability and support on the Linux and Intel OSX platforms. Due to recency of the Apple M1 chip there is no official support for this platform in bioconda. As such, we are not able to provide package manager support for that architecture yet. We have successfully built tn93 on M1 architectures from source and provide details for how to do so. A number of other commonly used packages also suffer from this problem and we will update our usability and support once an industry-wide solution is made available.

Comment #21

* There is no file named `snakemake_config.json` in the RASCL directory. The `README.md` configuration instructions should be corrected to refer to `config.json`.

Response 21: We have modified the README file to reflect these changes.

Comment #22

* The configuration instructions are unclear. If the "clade of interest" is set to `B.1.1.7`, does that refer to a FASTA filename (user input data), or to labels contained within that FASTA file? Why don't these instructions employ the same terminology as the manuscript, *i.e.*, "query" and "background"? In the file `cluster.json`, what do the labels `cluster`, `nodes`, `ppn` and `name` correspond to?

Response 22: We have modified our analysis configuration file (“config.json”) and the corresponding instructions to add clarity:

● “Clade of interest” now refers to the “Label” variable, which is used to annotate the phylogenetic tree.

● The “Query_WholeGenomeSeqs” refers to the relative location of the query whole genome dataset (e.g. “Example1/Query-Alpha.fasta”) 

● The “Background_WholeGenomeSeqs” refers to the relative location of the query whole genome dataset (e.g. “Example1/Background-preAlpha.fasta”) 

● All other variables include the same terminology for consistency: "max_background", "threshold_background", "max_query", "threshold_query" and are further explained in our README file.

We have modified our README file to contain explainer text about our cluster configuration file (“cluster.json”). 

● The “cluster” variable refers to the workload manager

● The “nodes” variable is a request for resource allocation from the server, in this case it refers to the number of nodes.

● The “ppn” variable is a request for resource allocation from the server, in this case it refers to the number of processors per node.

● The “name” variable is a specification to submit the jobs for the RASCL application to a specific queue. These have different names and priorities, please refer to your local system administrator for more information.

● We have added an additional variable “walltime” which is a request for a certain period of time for resource allocation from the server.

Comment #23

* Running RASCL appears to require the Slurm workload manager (`qsub`). If so, then this should be listed as a package dependency. However, users should have the option of running RASCL without a workload manager.

Response 23: We have provided a “run_Local.sh” version of the submitting script to facilitate the user running the program on a local machine without a workload manager.

Comment #24

* observablehq.com notebook has older GitHub link, https://github.com/veg/SARS-CoV-2_Clades

Response 24: We have modified the ObservableHQ notebook to reflect our latest link at https://github.com/veg/RASCL

Comment #25

* "Label tree with amino-acids" checkbox does not seem to be working.

Response 25: This checkbox is only active when viewing a specific site within the viral genome, for example in the “View this site (in SARS-CoV-2 reference coordinates)”. When viewing a single site, the phylogenetic tree viewer shows a site-level tree with the codon at that position for each sequence by default. When the checkbox is enabled, we translate the codon into its corresponding amino acid in the phylogenetic tree viewer. We have modified the checkbox label for clarity “Label tree with amino-acids (Site-level trees only)”. 

Comment #26

* My JS console is listing a lot of errors on page load that appear to be associated with phylotree.js; for example:

* Error: attribute d: Expected number, "MNaN,0LNaN,NaNLNa…".

* Error: attribute transform: Expected number, "translate (NaN,NaN) ".

Response 26: We have created an issue to change the console errors to warnings with the package phylotree.js developers.

Comment #27

* for the notebook interface, is it possible to provide some visual cue to indicate that table rows can be sorted by clicking on column labels?

Response 27: We have provided a visual cue to the table columns in the form of an up/down arrow to indicate that sorting is permissible.

Comment #28

* can the developers please provide an additional set of instructions for running RASCL in a Linux environment that is not running Anaconda? Some shared computing environments prohibit users from using Anaconda, e.g., Compute Canada.

Response 28: We have provided instructions in our README for non-conda based installation of environment dependencies.

Comment #29

* please provide instructions for retrieving JSON data from the observablehq site via some API.

Response 29: Observable notebooks allow retrieval of any named cells via embedding (https://observablehq.com/@observablehq/embeds) or data export from named cells. We believe this should be sufficient for most users.

Comment #30

* The Python scripts have some weird formatting. For example, in `tn93_cluster.py` there are long spaces within `add_argument()` calls, each on a single line. Please conform to PEP 8 conventions if possible. `generate-report.py` has a monolithic for-loop spanning over 500 lines. This code would be much more maintainable if the developers applied a more modular code style. Also see lines 689-695 and 743-749 for weird whitespace.

Response 30: We have modified our custom python scripts for styling and maintainability accordingly.

---

## [Decision Letter · Decision Letter 1]

9 Sep 2022

PONE-D-22-06403R1RASCL: Rapid Assessment of Selection in CLades through molecular sequence analysisPLOS ONE

Dear Dr. Lucaci,

Thank you for submitting your manuscript to PLOS ONE. After careful consideration, we feel that it has merit but does not fully meet PLOS ONE’s publication criteria as it currently stands. Therefore, we invite you to submit a revised version of the manuscript that addresses the points raised during the review process.

We look forward to receiving your revised manuscript.

Kind regards,

Vladimir Makarenkov

Academic Editor

PLOS ONE

Journal Requirements:

Reviewers' comments:

Reviewer's Responses to Questions

**Comments to the Author**

1. If the authors have adequately addressed your comments raised in a previous round of review and you feel that this manuscript is now acceptable for publication, you may indicate that here to bypass the “Comments to the Author” section, enter your conflict of interest statement in the “Confidential to Editor” section, and submit your "Accept" recommendation.

Reviewer #1: All comments have been addressed

Reviewer #2: (No Response)

2. Is the manuscript technically sound, and do the data support the conclusions?

Reviewer #1: Yes

Reviewer #2: Yes

3. Has the statistical analysis been performed appropriately and rigorously? 

Reviewer #1: Yes

Reviewer #2: Yes

4. Have the authors made all data underlying the findings in their manuscript fully available?

Reviewer #1: Yes

Reviewer #2: Yes

5. Is the manuscript presented in an intelligible fashion and written in standard English?

Reviewer #1: Yes

Reviewer #2: Yes

6. Review Comments to the Author

Reviewer #1: I appreciate the amount of work that has gone into revising the manuscript and the source code. The manuscript is greatly improved and I was able to run the code on my Linux workstation.

Reviewer #2: Lucaci and collaborators introduces a new piece of software, RASCL, that allows for rapid monitoring of the evolution of SARS-CoV-2 strains. RASCL combines a sequence mapping, clustering, as well as numerous phylogenetic analyses to produce reports about the evolutionary selective trends in a given dataset. I think that the paper is relevant and generally well-written.

I think that the authors should use recombination detection algorithms available for example in SimPlot++ (Samson et al., Bioinformatics 2022) to identify sequence change due to recombination.

Also, I think that the paper could benefit from a short discussion about possible origins of SARS-Cov-2. You could rely on the following references in this discussion: Boni, Maciej F., et al. "Evolutionary origins of the SARS-CoV-2 sarbecovirus lineage responsible for the COVID-19 pandemic." Nature microbiology 5.11 (2020): 1408-1417. Domingo JL. What we know and what we need to know about the origin of SARS-CoV-2. Environ Res. 2021;200:111785. Makarenkov, V., Mazoure, B., Rabusseau, G. et al. Horizontal gene transfer and recombination analysis of SARS-CoV-2 genes helps discover its close relatives and shed light on its origin. BMC Ecol Evo 21, 5 (2021).

7. PLOS authors have the option to publish the peer review history of their article (what does this mean?). If published, this will include your full peer review and any attached files.

Reviewer #1: No

Reviewer #2: No

---

## [Author Response · Author response to Decision Letter 1]

16 Sep 2022

September 14, 2022

Dr. Vladimir Makarenkov

Academic Editor

PLOS ONE

RE: Response to reviewers for: RASCL: RAPID ASSESSMENT OF SELECTION IN CLADES THROUGH MOLECULAR SEQUENCE ANALYSIS.

Thank you for the opportunity to submit a revised manuscript on “RASCL: RAPID ASSESSMENT OF SELECTION IN CLADES THROUGH MOLECULAR SEQUENCE ANALYSIS”. 

Please find attached our revised contribution that incorporates responses to the Editor and Reviewer comments. Each of the comments have been addressed and a detailed response is attached. Both a marked-up manuscript and a clean LaTeX version of the paper are included.

We have taken special consideration to answer the important points raised by the reviewer regarding SARS-CoV-2 origins and genetic recombination detection, a limitation of our applications current implementation which we aim to handle in future versions of RASCL and that will enable it to play a more significant role in the global pathogen monitoring ecosystem. 

Yours sincerely,

Alexander G. Lucaci, M.S.

Ph.D. Candidate in Bioinformatics

Institute for Genomics and Evolutionary Medicine (iGEM)

Temple University

Comment #1

I think that the authors should use recombination detection algorithms available for example in SimPlot++ (Samson et al., Bioinformatics 2022) to identify sequence change due to recombination.

Answer #1 We have modified our manuscript to discuss the use and role of recombination detection in our application.

Comment #2 

Also, I think that the paper could benefit from a short discussion about possible origins of SARS-Cov-2. You could rely on the following references in this discussion: Boni, Maciej F., et al. "Evolutionary origins of the SARS-CoV-2 sarbecovirus lineage responsible for the COVID-19 pandemic." Nature microbiology 5.11 (2020): 1408-1417. Domingo JL. What we know and what we need to know about the origin of SARS-CoV-2. Environ Res. 2021;200:111785. Makarenkov, V., Mazoure, B., Rabusseau, G. et al. Horizontal gene transfer and recombination analysis of SARS-CoV-2 genes helps discover its close relatives and shed light on its origin. BMC Ecol Evo 21, 5 (2021).

Answer #2 We have modified our manuscript to include a discussion on the possible origins of SARS-CoV-2.

---

## [Editor Report · Decision Letter 2]

20 Sep 2022

RASCL: Rapid Assessment of Selection in CLades through molecular sequence analysis

PONE-D-22-06403R2

Dear Dr. Alexander G Lucaci,

We’re pleased to inform you that your manuscript has been judged scientifically suitable for publication and will be formally accepted for publication once it meets all outstanding technical requirements.

Kind regards,

Vladimir Makarenkov

Academic Editor

PLOS ONE
---

## [Editor Report · Acceptance letter]

3 Oct 2022

PONE-D-22-06403R2 

RASCL: Rapid Assessment of Selection in CLades through molecular sequence analysis 

Dear Dr. Lucaci:

I'm pleased to inform you that your manuscript has been deemed suitable for publication in PLOS ONE. Congratulations! Your manuscript is now with our production department. 

Kind regards, 

on behalf of

Dr. Vladimir Makarenkov 

Academic Editor

PLOS ONE